Corrected: Author correction

# Induced unconventional superconductivity on the surface states of $Bi_2Te_3$ topological insulator

Sophie Charpentier[1], Luca Galletti [1], Gunta Kunakova[1,2], Riccardo Arpaia [1], Yuxin Song [1,3], Reza Baghdadi[1], Shu Min Wang[1,3], Alexei Kalaboukhov[1], Eva Olsson [4], Francesco Tafuri[5,6], Dmitry Golubev[7], Jacob Linder[8], Thilo Bauch[1] & Floriana Lombardi [1]

Topological superconductivity is central to a variety of novel phenomena involving the interplay between topologically ordered phases and broken-symmetry states. The key ingredient is an unconventional order parameter, with an orbital component containing a chiral $p_x + ip_y$ wave term. Here we present phase-sensitive measurements, based on the quantum interference in nanoscale Josephson junctions, realized by using $Bi_2Te_3$ topological insulator. We demonstrate that the induced superconductivity is unconventional and consistent with a sign-changing order parameter, such as a chiral $p_x + ip_y$ component. The magnetic field pattern of the junctions shows a dip at zero externally applied magnetic field, which is an incontrovertible signature of the simultaneous existence of 0 and $\pi$ coupling within the junction, inherent to a non trivial order parameter phase. The nano-textured morphology of the $Bi_2Te_3$ flakes, and the dramatic role played by thermal strain are the surprising key factors for the display of an unconventional induced order parameter.

[1] Department of Microtechnology and Nanoscience, Chalmers University of Technology, SE-41296 Göteborg, Sweden. [2] Institute of Chemical Physics, University of Latvia 19 Raina Boulevard, LV-1586 Riga, Latvia. [3] Shanghai Institute of Microsystem and Information Technology, Chinese Academy of Sciences, 865 Changning Road, Shanghai CN-200050 China. [4] Department of Applied Physics, Chalmers University of Technology, SE-41296 Göteborg, Sweden. [5] Dipartimento di Fisica E. Pancini, Università di Napoli Federico II, IT-80126 Napoli, Italy. [6] CNR-SPIN Institute of Superconductors, Innovative Materials and Devices, Napoli IT-80125, Italy. [7] Department of Applied Physics, Aalto University School of Science P.O. Box 13500, FI-00076 Aalto, Finland. [8] Department of Physics, QuSpin Center of Excellence, Norwegian University of Science and Technology, N-7491 Trondheim, Norway. S. Charpentier and L. Galletti contributed equally to this work. Correspondence and requests for materials should be addressed to F.L. (email: floriana.lombardi@chalmers.se)

The notion of superconducting materials with fully gapped order parameters, but that still support topologically protected surface states[1–5], including Majorana bound states[2,3], are currently subject to much attention. Such systems intertwine two key paradigms in condensed matter physics, namely spontaneous symmetry breaking and topology, which has been predicted to cause unusual quantum phenomena with even direct analogies to concepts in high-energy physics, such as axion couplings[6]. This novel physics has been recently predicted to occur also in systems where the superconducting phenomenon involves topological Dirac electrons and it can be induced by the proximity with a conventional superconductor.

Heterostructures involving a conventional superconductor (S) and an exotic conductor, represented by the surface of a three-dimensional (3D) topological insulator[7–11] (TI) and the edge states of two-dimensional quantum wells[12,13], are ideal systems to emulate topological superconductivity. In the pioneering work by Fu and Kane[1], the superconductivity induced in the surface states of a 3D TI is described, in a basis where the operators acquire a phase factor, by an order parameter (OP) that resembles a spinless chiral $p_x + ip_y$ ($p$) but does not break time-reversal symmetry. In a subsequent work, Tkatchov et al.[14] have derived, this time in a canonical operator basis, that the induced superconductivity has an orbital term of the type $p + s$-wave with the $p$ part, consisting of conventional chiral $p_x + ip_y$ and $p_x − ip_y$, that sums up with the $s$-wave term through Pauli matrices to form the total superconducting order parameter.

It is clear that the first step that must be taken to unveil the rich variety of the predicted phenomena related to topological superconductivity is to clearly demonstrate the unconventional nature of the OP, which identifies in a chiral $p_x + ip_y$ ($p$-wave) orbital term. In compounds like the $Sr_2RuO_4$ layered perovskite, for example, considered the archetype of a topological superconductor, the existence of a chiral $p_x + ip_y$ wave orbital order, with a nontrivial internal phase, is still not generally accepted[15]. The appearance of Majorana fermions and topological quantum computation are fundamentally encoded in the properties of such $p$-wave superconductivity.

Josephson interferometry emerges as a crucial tool to show key features of the superconductivity involving topological Dirac electrons, as for instance supercurrents induced in the helical edge states of HgTe/HgCdTe[12] and InAs/GaSb[13] 2D TI quantum wells. The helical metallic surface states of 3D TI, probed by Josephson interferometry have shown, up to now, a mostly conventional induced superconductivity. Here the Josephson transport has been demonstrated to be ballistic[8,10] and uniquely associated with the helical surface states[16] which rules out a contribution from the disordered bulk[8,10]. However, no other signatures have been observed. The reports on a skewed Josephson current phase relation could not be clearly discriminated if due to a conventional high transparency barrier in S-3DTI-S junctions[17] or to the peculiar bound state spectrum of the surface helical metal[11,14] possibly hosting Majorana fermions.

Our experiment is a decisive step along this line. We have realized hybrid devices based on a 3D TI, where the nanostructured morphology of the topological insulator flakes makes the measurement of the Josephson effect sensitive to the interface unconventional order parameter symmetry. By using the same path and notions previously employed to prove the $d$-wave OP in high critical temperature superconductors[18,19] (HTS) we have measured an unconventional magnetic field pattern of the Josephson current, with a dip at zero magnetic field. We demonstrate that this is the incontrovertible code of a Cooper pair tunneling process probing an OP with a non-trivial internal phase.

## Results

### Unconventional magnetic field patterns in Al-$Bi_2Te_3$-Al Josephson junctions.

Al-3DTI-Al Josephson junctions (Fig. 1a) have been realized using $Bi_2Te_3$ flakes from epitaxial thin films. By varying the interface transparency $D$, from a value close to 1 to the tunneling limit ($D \ll 1$) we have been able to show that the surface states of $Bi_2Te_3$ host an induced OP with a non trivial phase compatible with a chiral $p$-wave. Figure 1b shows the nanostructured morphology of the flakes. We observe

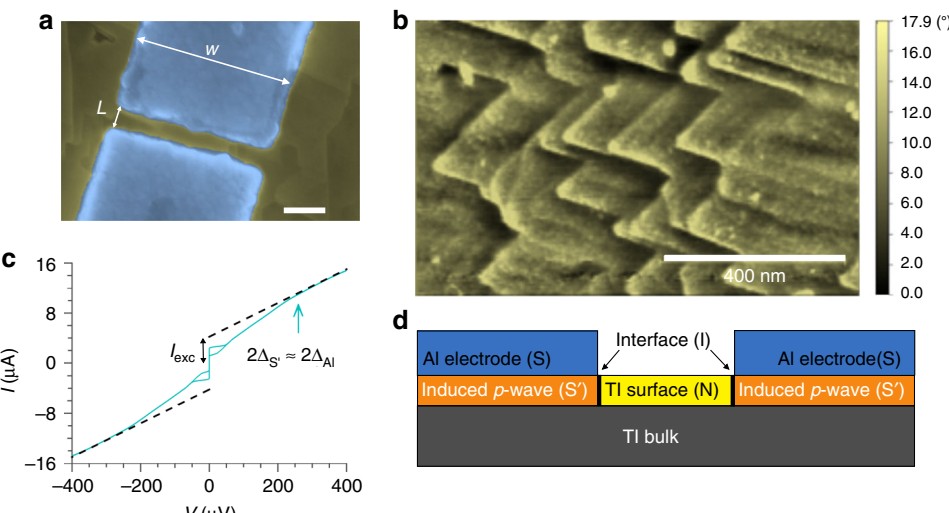

**Fig. 1** Morphological and transport characterization of a typical proximity based $Bi_2Te_3$ Josephson junction. **a** Scanning electron microscope image of a junction made using an exfoliated flake. The light blue areas define the Al/Pt electrodes while the $Bi_2Te_3$ flake is shown in yellow. Scale bar: 200 nm. **b** Atomic force microscope image of a $Bi_2Te_3$ thin film grown on GaAs substrate showing a typical growth with aligned pyramidal domains. Scale bar: 400 nm. **c** IVC of a typical junction ($w = 1 \mu m$, $L = 200$ nm) measured at 20 mK (cyan). The resistive branch shows a neat change of slope at a voltage $V \approx 260 \mu V$ (cyan arrow) that corresponds to $\sim 2\Delta_{Al}$ for the thickness of the electrodes[22]. The black arrow indicates the value of the extracted excess current. **d** Cross section schematics of the effective device under consideration. The transport properties can be assimilated to those of a S′INIS′ Josephson junction where S′ represents the induced superconductor in the $Bi_2Te_3$

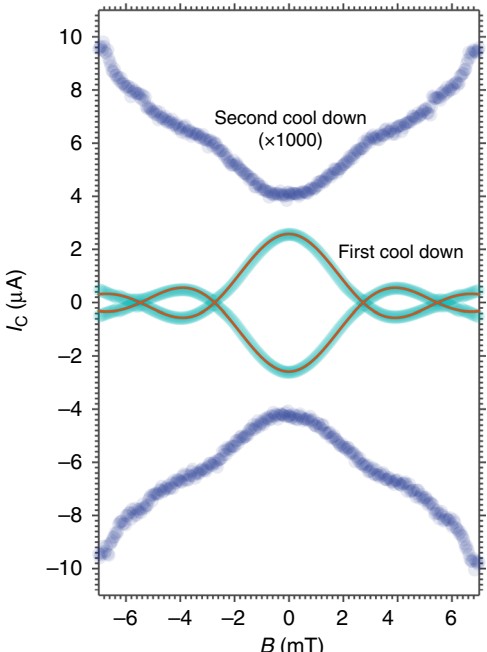

**Fig. 2** Magnetic field pattern of a typical junction before and after the thermal cycle. Critical current $I_C$ dependence on the externally applied magnetic field $B$ for the junction of Fig. 1c at 20 mK, before (cyan) and after (blue) the thermal cycle. At the first cool down the $I_C(B)$ dependence shows a conventional Fraunhofer dependence fitted with the expression $I_C(B) = I_C(0)|\sin(\pi\phi/\phi_0)/(\pi\phi/\phi_0)|$ (red line) where $I_C(0)$ is the critical current at zero field, $\phi_0$ is the flux quantum, and $\phi = BA_{eff}$ is the flux through the junction, being $A_{eff}$ the effective area. After the thermal cycle (blue points), the critical current is dramatically reduced (the data points are multiplied by a factor 1000 for clarity), and a dip at $B = 0$ appears

characteristic pyramidal domains (see Methods section) with a prevailing single domain, which points towards thin films without twin boundaries[20].

Figure 1c shows the current voltage characteristic (IVC) of a typical junction at the first cool down, measured at base temperature of 20 mK in a dilution refrigerator. The curve presents 50% hysteresis and a large excess current $I_{exc}$, defined as the extrapolation at zero voltage of the IVC above the Al gap, $\Delta_{Al}$. This value can be higher than the Josephson current for high transparent interfaces[21] as we generally find for our devices. We will assume that the junctions can be assimilated to S'INIS' structures where S' is the effective fully gapped $p + s$-wave superconductor[14], N is the $Bi_2Te_3$ Dirac metal channel and I the interface barrier (Fig. 1d). The induced gap in S' will define the maximum $I_C R_N$ product where $I_C$ represents the maximum Josephson current and $R_N$ the normal resistance of the junction. For the IVC of Fig. 1c, we get an $I_C R_N \approx 100\ \mu V$ indicating an induced gap close to that of the Al[22]. This is further supported by the high value of the transparency $D$ we extract from the IVC. Specifically from the value $eI_{exc}R_N/\Delta_{S'}$, with $\Delta_{S'} \approx \Delta_{Al}$, we can extrapolate the value of the barrier transparency $D \cong 0.8$[21].

Figure 2 shows the magnetic field dependence of the Josephson current for the same junction as in Fig. 1c. We observe an ideal Fraunhofer pattern with clear secondary lobes (cyan circles). From the modulation period $\Delta B = \phi_0/A_{eff} \approx 2.5$ mT, where $\phi_0$ is the superconducting flux quantum, we extract an effective area $A_{eff}$ of 0.75 $\mu m^2$. This value is very close (within few %) to the numerically calculated effective area, by assuming a $2\pi$ periodic current phase relation (Supplementary Note 1). This has been verified for several junctions on various chips, which confirms a

conventional magnetic field behavior of the devices in agreement with the theoretical predictions[23] and various experimental reports[24], while in contrast with the anomalies stated in earlier works[25].

After thermal cycling (from base temperature to 300 K and back to base temperature) the transport properties of all devices undergo a dramatic change. The critical current of the junctions is, for most devices, reduced by more than two orders of magnitude (Fig. 2) while the normal resistance increased by a factor between 1.1 and 6 (Table 1).

We shall argue that this phenomenology is related to the dramatic role played by strain to tune the interquintuple layers interaction and therefore the topological phase in TIs. First principle calculations[26,27] have shown that the consequence of a tensile strain (out of plane compression) is a shift of the Dirac point closer to the valence band while a compressive strain (out of plane expansion) leads to a gap opening at the Dirac point[26]. Strain is usually generated during the epitaxial growth of the material on the substrate, with lattice parameters different from those of the topological insulator. Recent reports have shown the tunability of the Dirac point by strain in thin topological crystalline insulator SnTe[28] and at grain boundaries in $Bi_2Se_3$ thin films[29].

Our experiment is quite different: exfoliated $Bi_2Te_3$ flakes are transferred to a $SiO_2/Si$ substrate, so a possible strain-related phenomenology cannot be attributed to the growth process. The strain instead is related to the huge difference in the thermal expansion coefficient of $Bi_2Te_3$ (~13.4 × 10⁻⁶ °C⁻¹) and that of the $SiO_2/Si$ substrate (0.5 × 10⁻⁶ °C⁻¹/2.4 × 10⁻⁶ °C⁻¹). The flake will experience this difference by the clamping to the substrate, through the patterning of the Al electrodes forming a nanometer sized gap junction. During the warming up of the sample the $Bi_2Te_3$ flake at the nanogap undergoes a compressive strain inducing plastic deformation that leads to a buckling of the $Bi_2Te_3$ channel forming the nanogap upon a subsequent cool down[30]. Figure 3 shows a SEM picture of a junction after cycling; a typical buckling feature appears in the nanogap (Supplementary Note 3).

Plastic deformations at the nanochannel can substantially affect the Josephson properties. The opening of a gap at the Dirac node, which can eventually reach the bulk gap[26], works as a tunneling barrier reducing the Josephson current as we observe in our experiment. The physics behind this reduction is that the opening of a gap at the Dirac node causes current-carrying surface states beneath the Al electrodes to become evanescent waves in the TI nanogap region, which thus decay over a shorter distance than in the gapless case and thus reduces the critical current.

For the devices, which undergo the most dramatic changes in the Josephson current, the value of the $I_{exc}$ is zero, signifying a very low transparency. However for these samples the most striking experimental observation is the inverted Josephson magnetic field pattern that appears after thermal cycling (Fig. 2 (blue points) and Fig. 4a, see also Supplementary Fig. 2, and Supplementary Note 2).

## Discussion

Our results are consistent with a proximity-induced superconductivity in the TI surface states compatible with a $p + s$ OP which contains a chiral p-wave term. Let us now focus on the effect of the interface transparency on the chiral p-wave part of the $p + s$ OP. Self-consistent calculations have shown that the transparency of the interface can have a dramatic effect on a chiral $p_x + ip_y$ wave OP. For highly transparent interfaces, both the $p_x$ and $p_y$ component of OP gap are equally suppressed at the

**Table 1 Relevant parameters for all the measured devices before and after the thermal cycle**

| Device | $I_C^{before}$ (μA) | $I_C^{after}$ (μA) | $R_N^{before}$ (Ω) | $R_N^{after}$ (Ω) | $I_C R_N^{before}$ (μeV) | $I_C R_N^{after}$ (μeV) | $w$ (μm) | $L$ (nm) | $\frac{I_C^{after}}{I_C^{before}}$ | $\frac{R_N^{after}}{R_N^{before}}$ |
|---|---|---|---|---|---|---|---|---|---|---|
| JM10 | 2.6 | 0.002 | 36 | 215 | 89 | 0.4 | 1 | 200 | 0.07% | 6.0 |
| JM5 | — | 0.017 | — | 216 | — | 3.6 | 0.5 | 100 | — | — |
| JM11 | 1.8 | 0.085 | 41 | 62 | 47 | 5 | 1 | 300 | 4% | 1.5 |
| SF2 | 0.58 | 0 | 15 | 20 | 9 | 0 | 2 | 100 | 0% | 1.3 |
| JM1 | 0.6 | — | 86 | — | 54 | — | 0.5 | 200 | — | — |
| SM4 | 9.3 | 0.060 | 17 | 61 | 144 | 8.6 | 1.5 | 100 | 0.6% | 3.6 |
| SD3 | 0.36 | 0.040 | 18 | 53 | 6 | 2 | 2 | 250 | 11% | 2.9 |
| SF3 | 2.2 | 0.030 | 10 | 25 | 22 | 0.7 | 2 | 100 | 1.4% | 2.5 |
| SF8 | 3.2 | 0.180 | 11 | 25 | 35 | 4.5 | 2 | 100 | 5.6% | 2.3 |
| SD4 | 0.6 | 0.02 | 32 | 52 (59) | 2 | 1 | 2 | 250 | 3% | 1.6 |
| SF4 | 19.8 | 6.600 | 3.6 | 5.6 | — | — | 6 | 100 | 33% | 1.5 |
| SD2 | 0.52 | 0.012 | 17 | 20 (73) | 9 | 0.2 | 4 | 250 | 2.3% | 1.2 |
| SD1 | 1.2 | 0.086 | 27 | 30 | 32 | 2.5 | 2 | 250 | 7.1% | 1.1 |
| SM1 | 11.0 | — | 15 | — | 151 | — | 1.5 | 100 | — | — |

The critical current $I_C$, the normal resistance (measured above gap) $R_N$, the $I_C R_N$ product, the width of the device $w$ (in case of a SQUID twice the junction) and the distance between the electrodes $L$ are reported. Devices JM10, JM5, SM4, SF8, and SD1 showed inverted magnetic patterns in the second cool down. The devices JM10 and JM5 are discussed in the text. The junctions start with a J in the device name, whereas the SQUIDs device names start with an S

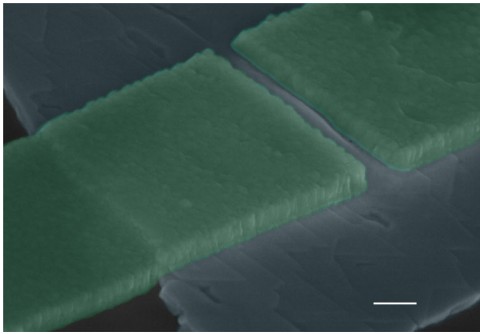

**Fig. 3** Colored SEM image of a typical junction after the second cool down. The picture (scale bar: 200 nm) shows a junction presenting an inverted Josephson magnetic pattern, after the second cool down. A clear buckling feature, induced by a compressive strain, is visible in the nanogap

interface[31] (Fig. 5a); however the net pairing symmetry of the order parameter remains a chiral $p$-wave at the interface region, with a slightly reduced magnitude. As the interface transparency is lowered, the $p_x$ component of the order parameter will be strongly suppressed, while the $p_y$ slightly enhanced[31,32] (Fig. 5b). Considering our data at the first cool down the transparency is high; if the induced order parameter is of the type $p + s$ it will preserve the same symmetry at the interface. In this case, a Fraunhofer-like magnetic pattern is expected[23], as we have found in our experiment. After thermal cycling, however, the critical current is suppressed by orders of magnitude (Fig. 2). This fact cannot solely be attributed to a change in interface transparency due to plastic deformations induced by compressive strain (buckling). The strong suppression of $I_C$ is, on the other hand, fully consistent with a change in pairing symmetry of the $p + s$ OP. In particular, as theoretically predicted[31,32] the chiral $p$-wave term will assume a predominantly $p_y$-wave form at the interfaces (with the $p_y$ lobe parallel to the interface).

If one considers only the chiral term of the $p + s$ OP its amplitude and phase at the S′IN interface are shown in Fig. 5c and 5d for a highly and low transparent interface respectively[32]. The chiral $p$ term, of the total OP, is mainly transformed in $p_y$ for low transparency barriers. Theory shows that in $p_y$-$p_y$ junctions the critical current becomes strongly suppressed even for a slight increase of scattering in the junction[33]. This is therefore

consistent with the observed reduction, of orders of magnitude in $I_C$, after thermal cycling. Secondly, such a change in pairing symmetry also explains the inversion of the Fraunhofer pattern with a dip at $B = 0$ when the S′IN interface transparency is lowered. For a predominantly $p_y$-wave OP at the interface, the occurrence of scattering in the normal region of the junction will couple quasiparticle trajectories, emerging from positive $p_y$ lobe of the OP on one side of the junction, with trajectories that probe the negative $p_y$ lobe on the other side of the junction. This corresponds to a net $\pi$-phase shift in the Josephson coupling (Fig. 6a and b). Here it is worth pointing out that while the buckling wave in the nanogap area can partially be responsible for the reduction of the Josephson current (and possibly for a non homogenous distribution of $J_C$ across the width of the junction) it cannot account for the peculiar dip of the Fraunhofer patter at $B = 0$, requiring a net $\pi$-phase shift between the electrodes. While the above scenario provides a possible route for introducing 0- and $\pi$-trajectories in the system, we underline that an s-wave component of the OP simultaneously exists. We speculate that for $\pi$-shifted trajectories to become possible, it might be of importance that the $p_y$-wave component is enhanced compared to the s-wave one since the $p$- and s-wave condensates are not separated due to the spin-mixing resulting from the normal-state TI dispersion with spin momentum locking.

In films with pronounced pyramidal nano-domains, like the flakes used in our experiment, the edges have proven to host an enhanced density of states[34,35] and to be preferential trajectories[36]. Aharonov-Bohm oscillations in the magneto-resistance of thin films, where orbits around the triangle edges define the loop areas[36] have been recently detected, demonstrating that the triangle's corners and more in general step-edges work as scattering centers. To show that this effect is also observed in our $Bi_2Te_3$ flakes we have measured the magneto-transport down to 20 mK temperatures. We observe clear periodic magneto-resistance oscillations, arising from coherent scattering of electron waves from the corners of the pyramidal domains and/or from the points where two domains merge, with a period corresponding to the morphology of the flake (Supplementary Note 4). In this scenario, among the quasi-particles crossing the $Bi_2Te_3$ normal channel, the ones undergoing large scattering angles at the corners or step-edges of the pyramidal nano-domain, will probe a phase shift of $\pi$ on the other side of the junction. The quasi-particles trajectories, instead, that do not undergo scattering events in the normal channel (e.g., coupling

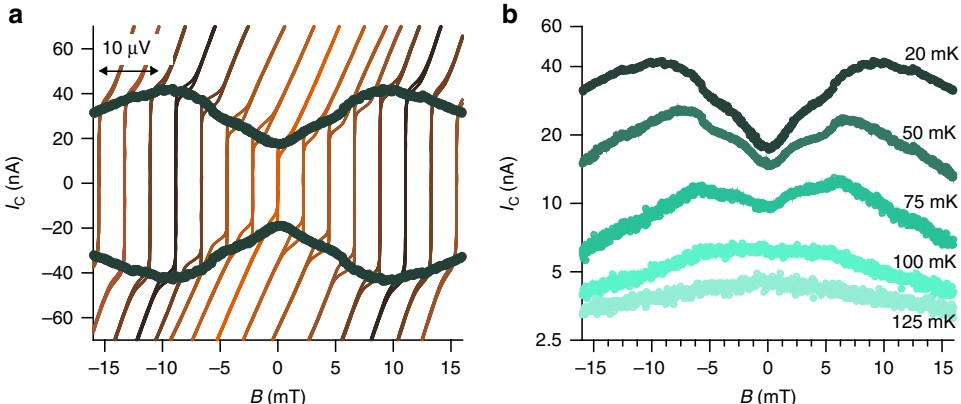

**Fig. 4** Current voltage characteristics as a function of magnetic field and temperature for a typical junction after thermal cycling. **a** IVCs of a second device ($w = 500$ nm and $L = 100$ nm) are shown as a function of $B$ after thermal cycling. The curves are shifted in voltage by a value proportional to the magnetic field. The green points represent the magnetic pattern of the junctions obtained using a voltage criterium $V = 600$ nV. Also in this case the magnetic pattern presents a very pronounced dip at $B = 0$. **b** Evolution of the magnetic pattern shown in **a** as a function of the temperature. The dip flattens out at a temperature close to 100 mK

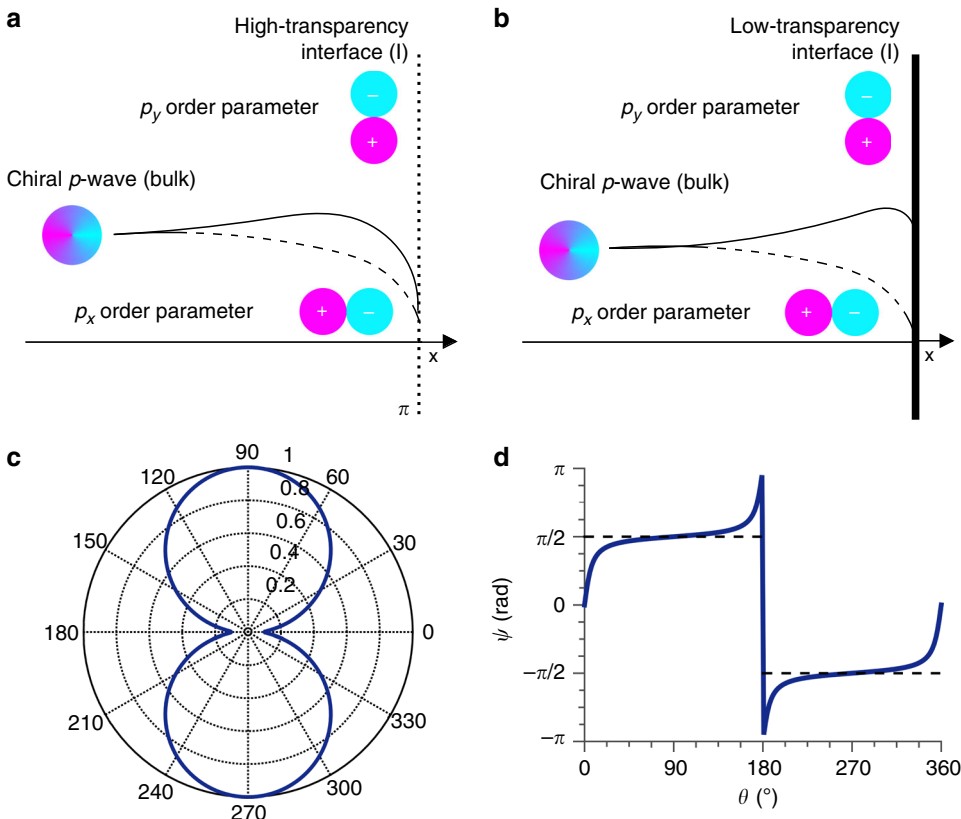

**Fig. 5** Proximity-induced OP on the surface of Bi$_2$Te$_3$. We consider only the chiral $p_x + ip_y$ part of the total $p + s$ OP. Close to the interface I the chiral term is modified: the $p_x$ (dashed line) and $p_y$ (straight line) components exhibit different magnitude. In case of high transparency interface (**a**), the two components are both suppressed in equal manner and the resulting interface OP still preserve the $p_x + ip_y$ wave character. For a low transparency interface (**b**), the $p_x$ component is highly suppressed while the $p_y$ slightly enhanced. The resulting OP has a predominantly $p_y$ character. **c** shows the amplitude of the resulting interface OP by considering a mixture $p_y + i\alpha p_x$, with $\alpha = 0.1$. This corresponds to the reduction of $p_x$ on distances of the order of the coherence length of S'[28]. The corresponding phase is shown in **d**

lobes of the same sign with each other) give a conventional 0 coupling. The schematics of the scattering processes are shown in Fig. 6b. The net result is that the total supercurrent is partially canceled by these competing contributions, which amounts to a suppressed $I_C$ at zero field.

This simple explanation has also a direct microscopic analog in terms of Andreev levels mediated transport in a Josephson junction[37]. Indeed a peculiarity of our measurement is the transition of the magnetic pattern to a conventional Fraunhofer type, for temperatures around 100 mK when the dip at zero field

**a**

**b**

**Fig. 6** Sketch of the device showing the induced order parameter and the occurrence of π-paths across the junction. **a** Three-dimensional sketch of the $Bi_2Te_3$ device. The Al electrodes (S) induce an effective $p + s$-wave superconductivity on the surface of the TI. In the picture we consider only the chiral term of the total OP. In close proximity of a low transparency interface the chiral $p_x + ip_y$ changes symmetry resulting in a predominant $p_y$ component (see main text for details). **b** Top view of the device. The occurrence of scattering at the pyramid corners and/or at the merging points of two pyramidal domains, in the N region of the junction, makes quasiparticle trajectories emerging from a negative $p_y$ lobe (blue arrow) on one side of the junction to couple with trajectories that probe the negative $p_y$ lobe (red arrow) on the other side of the junction. In case of scattering-free transport in the N region the quasi-particles trajectories probe the same phase in both electrodes (red arrows)

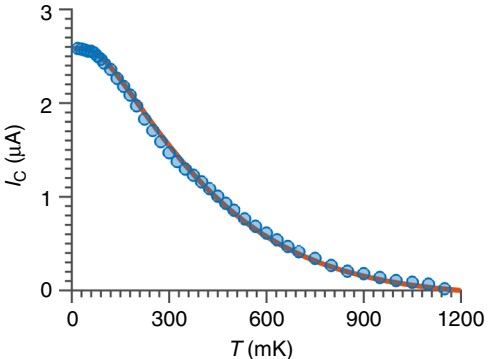

**Fig. 7** Temperature dependence of the Josephson current for a high transparency junction. Critical current as a function of the temperature, measured at the first cool down for the junction of Fig. 1c (full circles). The red line is the fitting considering a quasi-ballistic transport model (see Supplementary Note 6 and Supplementary Equations (5–10)). We extract the following fitting parameters for the mean free path $l_e = 130$ nm and the transparencies of the two barriers $D_1 = D_2 = 0.98$. From Supplementary Equation (13) the expected normal-state resistance is $R_N = 70 \, \Omega$. The experimental value for $R_N$ is instead $35 \, \Omega$. The discrepancy between these two values can be attributed to the shunting effect of the bulk of the $Bi_2Te_3$ flake

flattens out (Fig. 4b). This crossover has been theoretically predicted in the framework of low transparent Josephson junctions involving d-wave HTS[37,38]. The physics behind this phenomenology is connected to the presence of zero energy Andreev bound states, or mid gap states (MGS), characteristic of Josephson junctions with an OP with a nontrivial internal phase[37]. For the case of Fig. 5b, the $p_x$ component, responsible of the MGS formation[31], is non zero on distances where Andreev reflection takes place[37] (which are given by the coherence length in S′). Mid gap states are therefore formed on both sides of the junction and their resonant coupling gives an additional Josephson transport channel proportional to $\sqrt{D}$ (instead of $D$ as for conventional Andreev level mediated transport). When the MGS carrying current dominates at very low temperature[37,38] and scattering events take place at triangle corners, the equilibrium phase difference of the system is at π, corresponding to a negative current. For straight quasiparticle trajectories the MGS current gives a conventional 0 phase character ground state.

Formally, this scenario is similar to artificially designed 0-π ferromagnetic Josephson junctions where multiple connected

segments are π-shifted compared to each other[39]. In this case, a similar inverted pattern is observed. The dip of the Josephson current at $B = 0$ is thus the proof of a superconducting order parameter with an internal phase-structure, which can cause either 0 or π couplings, precisely as seen in our setup (Supplementary Note 5).

In a generic junction, the contribution to the current is due to both the resonant MGS and the conventional Andreev bound states. At larger temperature the contributions of MGSs current cancels out since both MGSs carrying forward and backwards current are populated[37] and therefore the conventional Andreev levels current will dominate. This leads to a peculiar π to 0 transition by increasing the temperature. Signatures of such a crossover have been reported in a superconducting quantum interference device (SQUID) geometry for d-wave junctions[40]. However those results were never confirmed, posing serious doubts on the effective possibility to detect a π−0 transition from the temperature dependence of the magnetic patterns. Our experiment is therefore one of the first demonstrations of Josephson transport mediated by MGS.

Clearly the dip at $B = 0$ in the magnetic pattern is also compatible with a $d_{x^2-y^2}$ wave order parameter[41]. We can rule out this possibility since the $d_{x^2-y^2}$ OP does not change symmetry depending on the interface transparency[37] and therefore cannot account for the dramatic modifications of the magnetic field response of the junctions upon thermal cycling. An helical $p_x + p_y$ OP can instead be compatible with our entire experimental scenario. However, this OP symmetry does not come directly from the theoretical works[1,14,41], so the possibility to induce an helical $p_x + p_y$ OP needs further theoretical assessment.

To further check the conjecture that step-edges and pyramidal corners act as scattering centers, we have fitted the temperature dependence of the critical current $I_C(T)$ of the high transparency junctions. We consider that our S′INIS′ junction is characterized by insulating barriers with transparencies $D_1$ and $D_2$, which are separated by a distance $L$[42]. The disorder in the normal channel is characterized by a mean free path $l_e$. As previously discussed, in case of high transparency barriers the junction behaves similarly to conventional s-wave junctions[43]; in this case we do not lose generality by modeling the S′ superconductor with an s-wave OP. The Josephson current for an S′INIS′ junction in the clean limit $l_e \gg L$ has been derived in a previous work[8]. To fit our data, we have modified the original result of ref. [8] in two ways (Supplementary Note 6): (i) by adapting it to the two-dimensional Dirac character of the surface states, and (ii) by extending it into the regime $l_e \approx L$ which is achieved by changing the flight time

expression of the electrons between the leads. We have verified that our approach well reproduces the Eilenberger theory for purely ballistic junctions and the Usadel theory for short diffusive junctions. Figure 7 shows the $I_C(T)$ of the junction of Fig. 1c. The solid curve is the fit by considering the geometrical dimensions and $l_e = 130$ nm. The extracted value for the mean free path clearly supports our assumption that few scattering centers are present in the barrier and that some quasi-particles while crossing the TI channel will undergo a single scattering event. The value of the normal-state resistance extracted from the fit is higher than the experimental one, which can be explained by shunting effect of the bulk of the $Bi_2Te_3$[8] (Supplementary Note 6).

It is worth mentioning that we did not observe any apparent correlation between the orientation of the nano-pyramidal domains and that of the electrodes. This is not surprising; the basic mechanism to get an inverted magnetic field pattern, is the occurrence of 0 and $\pi$ trajectories within the nanogap and they can be obtained for whatever orientation of the triangles with respect of the electrode, provided that some of the corners and/or the points where two domains merge fall within the $Bi_2Te_3$ nanogap (Supplementary Note 4).

Finally we would like to point out that magneto transport measurements (Supplementary Note 4) have clearly shown that scattering mechanisms, connected to the morphology of the $Bi_2Te_3$ flakes, take place in the junction channel, which is instrumental to get $\pi$ trajectories. However the interpretation of our measurements does not change if other scattering mechanisms are also involved.

To conclude it is worth discussing if the interpretation we have provided of our experiment would change if we would also consider a parallel Josephson contribution due to the transport through the bulk with an s-wave OP. Cooper pair transport through the bulk would just act as 0 trajectories effectively lifting the value of the Josephson current at zero external magnetic field, however leaving unaltered the dip structure at zero field of the magnetic pattern.

## Methods

**Materials and device fabrication.** $Bi_2Te_3$ thin films have been deposited by molecular beam epitaxy on a GaAs (100) substrate with a 2° vicinal cut. Prior to the growth, a 1 min Te soaking to the GaAs surface was carried out for passivation purpose. Then, the growth started at a temperature of 180 °C, when both Te and Bi sources were opened, and ended at a thickness of 80 nm after 1 h. The films have been grown with a high Te/Bi flux ratio, to obtain high quality crystallinity, as verified by X-ray diffraction (XRD) analysis[44].

The films show characteristic aligned pyramidal domains (Fig. 1b). This morphology is different with respect to the growth on exact substrates, where two types of pyramidal domains rotated by 60° with respect to each other are observed[20]. As verified by XRD analysis, our films show 86% single domain, which indicates that the vicinal cut in the substrate leads to suppression of twin boundaries, as also reported in literature[20]. The transport properties have been characterized by magneto-transport measurements; the films show an effective surface carrier density of $4 \times 10^{13}$ cm$^{-2}$ and mobility above 2000 cm$^2$ V$^{-1}$ s$^{-1}$ (at the temperature of $T = 2$ K).

Josephson junctions using Al electrodes have been realized by transferring the flakes on a Si/SiO$_2$ substrate. To improve the adhesion with the Al electrodes a thin layer (3 nm) of Pt is deposited in situ on the surface of the flake, previously cleaned by a short (10 s) Ar$^+$ ion milling[45,46].

**Data availability**. All relevant data are available from the authors.

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

## Acknowledgements

The work has been supported by the Knut and Alice Wallenberg Foundation under the project "Dirac Materials." We thank J. Kirtley, A. Leggett, S. Kubatkin, P. Lucignano, A. Tagliacozzo, M. Fogelström, M. Eschrig, P. Burset, V. Shumeiko, and A. Black-Shaffer for inspiring discussions. This work was partly supported by the Research Council of Norway through its Centres of Excellence funding scheme, project number 262633, QuSpin.

## Author contributions

S.C. and L.G. have fabricated the samples and performed the measurements. S.W. and Y.S have grown the $Bi_2Te_3$ thin films. D.G. and J.L. have contributed with theoretical insights and F.T. with discussions. G.K., E.O., A.K., R.A., and R.B. have performed morphological and structural characterizations of the devices. F.L. and T.B. have interpreted the measurements in collaboration with all authors. F.L has written the paper with inputs from all co-workers.
