## [Peer Review File · Nature Communications]

Editorial Note: This paper has been previously reviewed at a journal not currently engaging in a Transparent Peer Review scheme. This file includes the reviewer comments and author responses while at Nature Communications.

Reviewers' comments:

Reviewer #4 (Remarks to the Author):

The manuscript of Charpentier et al. reports on the observation of a supercurrent minimal at zero magnetic field in a topological insulator-based Josephson junction. They attribute this minimal to be the co-existence of 0 and π -coupling channels in the Josephson junction, which is further due to the strain and low contact transparency after device thermal cycling. The authors then discussed the origin of this π -junction, and find it to be consistent with a non-trivial topological phase in the junction.

Inducing superconductivity in topological insulator surface can give effective p-wave superconductivity and Majorana bound states, which is one of the hottest topics in condensed matter. One common method to study this non-trivial topological phase is through Josephson interferometry, i.e. magnetic field dependence of supercurrent in an S-3D TI-S Josephson junction system. Although many experimental studies in this system have shown some unconventional interference patterns (e.g. skewed current-phase relation). Most of the studies suffer from their 3D topological insulator not being ideal (e.g. co-existence of bulk conduction, trivial edge-states, etc). This makes the claim of topological superconductivity, to some extent, controversial.

In my opinion, this work has unambiguously demonstrated a new phenomenon in these types of systems, i.e. supercurrent minimal at zero magnetic field in an S-TI-S junction. This is an impressive observation. Even though it will be better to have this observation in devices which are NOT thermal cycled (because the buckling by thermal cycling in the Josephson junction adds some uncertainty during the interpretation of the data). Based on the previous referee report and the authors' comment, there is no obvious trivial alternatives which can explain this effect. So I think the claim of existence of π -junction is, to some extent, convincing. However the argument to relates this phenomena to non-trivial topological phase seems speculative for me. But considering the novel observation of supercurrent minimal and potential impact of this work if the claims turned out to be right, I think this work deserves a publication in Nature Communications.

Reviewer #5 (Remarks to the Author):

In the manuscript titled "Topological induced superconductivity on the surface states of Bi₂Te₃" by S. Charpentier, et al., the authors study Al-Bi₂Te₃-Al Josephson junctions. They report the temperature dependence of the critical current, and the critical current diffraction patterns with applied magnetic field of such devices. Consecutive thermal cycles lead to different junction characteristics: First, large $I_c R_n$ -products, excess current, and a Fraunhofer-like diffraction pattern are observed. After the first thermal cycle, the critical current of all devices drops by (several) orders of magnitude. Additionally, about one third of the devices exhibit a qualitative change in the diffraction pattern, viz., a dip in critical current at zero field. SEM and AFM images show that the topological insulator (TI) material between the Al electrodes buckles when cycled thermally due to the different thermal expansion coefficients of substrate, superconductor, and TI. To explain the unusual transport behavior, the authors suggest that scattering in the weak link allows electrons to experience a π phase shift by tunneling into different lobes of the effective p-wave superconducting order parameter presumed to exist in the TI material underneath the Al electrodes. This is expected to create an array of randomly

distributed channels (or domains) with "0" and "pi" phase shift, hence the dip at zero magnetic field.

I find the observation intriguing, however, the explanation by the authors is not convincing. It bases on a series of assumptions, some of which seem implausible or unclear to me:

(a) It is widely believed that processes such as Ar ion milling create additional defects and thus increase (electronic) doping of the affected area. Also, the presence of a metallic film will likely cause band bending in the material underneath the electrodes. Therefore, I would expect some bulk doping in the S' region of the device. Does it make sense to assume that the induced order parameter has p-wave symmetry? A scenario with perhaps mixed effective order parameter (s- and p-wave) seems more likely to me. Does the proposed mechanism of creating an effective p-wave order parameter after thermal cycling work when the bulk conducts?

(b) The authors assert that, a low transparency barrier forms because mechanical strain opens a gap in the band structure. It would appear to me that buckling removes some of the strain. Also, when I look at Fig. S3, I expect the strain profile to be not homogeneous along the width of the junctions and thus the transmission and the critical current density to vary accordingly. Can this be reconciled with the modeling assumptions in Fig. S11 (b) of the Supplemental Materials, i.e., many channels (or domains) with a similar magnitude of the critical current density? Could one obtain diffraction patterns in the shape of Figs. S2 and S11 with a non-uniform critical current distribution in the junction?

(c) It seems that the suggested mechanism for creating pi channels will work only if transport remains quasiballistic after mechanical deformation (or few scattering events occur in the weak link region). However, I would expect strong mechanical deformation to add crystal defects and thus scattering centers. Have the authors thought about this possibility?

(d) Can we simply assume that the interface transmission between the metal and the TI remains unchanged when the material deforms? Does the strain relax underneath the metallic electrodes? Is there no change in the band structure properties of the TI in the S' region?

(e) It is also unclear to me why

Referee#5

The authors assert that, a low transparency barrier forms because mechanical strain opens a gap in the band structure. It would appear to me that buckling removes some of the strain. Also, when I look at Fig. S3, I expect the strain profile to be not homogeneous along the width of the junctions and thus the transmission and the critical current density to vary accordingly. Can this be reconciled with the modeling assumptions in Fig. S11 (b) of the Supplemental Materials, i.e., many channels (or domains) with a similar magnitude of the critical current density? Could one obtain diffraction patterns in the shape of Figs. S2 and S11 with a non-uniform critical current distribution in the junction?

The referee is right: the strain profile is not homogeneous along the junction width. This is clear from Supplementary Fig. 3d showing an AFM line scan of the buckling morphology along the junction width, where one can clearly identify peaks and valleys. We have explicitly mentioned this structural feature in the text of the supplementary information. In a recent work by the group of Vydia Madhavan (Urbana Champaign) [ref. 28 of our manuscript, *Nat. Nanotechnol.* **10**, 849-853, doi:10.1038/nnano.2015.177 (2015)] a 2D periodic buckling structure has been induced in SnTe Topological Crystalline Insulator through a PbSe substrate with a large lattice constant mismatch. STM measurements of the lattice constant in these thin films have been able to identify different strain in the elevated topographic areas (tensile) compared to the valley topographic areas (compressive). In our study we are dealing with a completely different system Bi_2Te_3 flakes anchored to the SiO_2/Si substrate through the Al electrodes. However because of the analogies between the morphologies of the buckling in the two experiments (ours and that of ref [28]), it is plausible to assume that also in our case the peaks and the valleys of the buckling structure have different strain. In particular for the regions of a compressive strain in Bi-based 3D TIs there is an opening of the gap at the Dirac node.

As suggested by the referee it is therefore reasonable to assume that the local critical Josephson current density is non homogenous across the width of the junction. However this does not change the main outcome of our paper: the minimum of the Josephson current magnetic pattern at zero external field can only be obtained if "0" and " π " path are present along the junction width, no matter if the absolute

Fig. 1: (a) Critical current density along the junction. The regions with negative current density are π -facets. (b) Critical current as a function of applied magnetic flux for the critical current density shown in (a). (c) Fraunhofer pattern of a regular junction.

value of the local critical Josephson current density is varying along the junction. This is because the appearance of the minimum in the magnetic pattern at zero field is exclusively due to the " π " phase changes along the Junction width. The amplitude of the local critical current density does not have any role.

We have simulated various patterns with random "0" and " π " facets, along the junction, characterized by different values of the local critical Josephson current density (here the π facets are characterized by a negative critical current density). An example is reported in Fig. 1. As for the case of uniform distribution of Supplementary Fig.11, we can clearly identify changes of slopes at Φ_0 and $2\Phi_0$ and $3\Phi_0$ (indicated by the orange lines), which correspond to the periodicity determined by the width of the junction. Here it is worth pointing out that the resemblance with our experimental data can be

improved by running our code hundreds of times, to find the random distributions of facets which best reproduce our data (as we did for the case of Supplementary Fig.11). At this point we do not think it is worth the effort since we clearly see that the minimum of the magnetic

pattern at zero external field can be reproduced both with uniform and non-uniform current distribution across the facets further supporting our statements that we have performed a “phase sensitive experiment”. To stress this point we have added the following sentence at the end of paragraph IV of supplementary material:

“It is worth pointing out that one can obtain a similar pattern as the blue curve of Supplementary Fig.11 (b) also by considering a non uniform Josephson current distribution among the “0” and “ π ” facets”

Referee #5

It seems that the suggested mechanism for creating pi channels will work only if transport remains quasiballistic after mechanical deformation (or few scattering events occur in the weak link region). However, I would expect strong mechanical deformation to add crystal defects and thus scattering centers. Have the authors thought about this possibility?

We thank the referee for mentioning this possibility. However we see no obvious reason for why the presence of multiple scattering centers, which strongly scatter the quasiparticles, would destroy the π channels - in fact, the scattering centers are the reason for why the π -channels exist in our proposed mechanism.

Referee #5

Can we simply assume that the interface transmission between the metal and the TI remains unchanged when the material deforms?

The thermal expansion coefficient of Al ($\sim 23 \times 10^{-6}/^{\circ}\text{C}$) is similar to that of the Bi_2Te_3 . For a good bonding between the Al and the flake, as the one provided by the Pt sticking layer, one can consider the Al electrodes and the flake under them as forming a quite homogenous material with a thickness more than twice that of the flake (the thickness of Al is roughly twice that of the flake). Because of this fact we do not expect that strain will affect the interface between the Al and the flake.

Referee #5

Does the strain relax underneath the metallic electrodes? Is there no change in the band structure properties of the TI in the S' region?

An exhaustive answer to these questions would require a rigorous strain analysis and therefore mechanical modeling of our devices. This is a task of paramount difficulty because of the many parameters simultaneously at play involving several factors like the geometry of the flake, that of the electrodes and their positioning with respect to the flake, the nanogap width and length and above all of the crucial details of the interface Al/ Pt/flake. Phenomenological considerations brings to reasonably assume that the strain is located at the nanogap. This is because the Al and flake underneath are in rigid contract and expand together applying strain to the nanogap.

AFM inspection of the Al electrodes do not show any wavy morphology, indicating that the part of the flake underneath has not undergone any structural change. The same is true also for the flake around the nanogap: the waving of the buckling exponentially dies out moving from the nanogap region. This is typical for a 3D strain phenomenon where the perturbation remains local and supports the fact that Al+ flake underneath behave like an homogeneous material.

However if some strain is left in the part of the flake under the electrodes this will not change the interpretation of our measurements. A possible residual tensile strain will move the Dirac point towards the valence band making the TI under the electrode only more doped. If a compressive strain remains, instead, there will be a gap opening at the Dirac point, the induced OP will lose the p-wave component and everything will be conventional. This is against our findings. A possible mixture of compressive and tensile strain could simply make the induced superconductivity non homogeneous, which could be indistinguishable from a non homogeneous Josephson current due to buckling (as also suggested by the referee#5)

Referee #5

It is also unclear to me why the manuscript describes a particular scattering mechanism in great detail and then concludes that, indeed, any scattering mechanism should lead to a similar effect.

The referee is completely right on this issue. The reason for the emphasis on the scattering mechanism from the corner of the pyramidal domain and /or the joint points between two pyramidal islands, is due to the many questions of referee#2. He explicitly asked

us to give experimental evidences showing that scattering events indeed were taking place inside the Bi_2Te_3 channel. His requests have required an enormous additional experimental work. Referee#2 explicitly asked us to show that Aharonov-Bohm (AB) oscillations, associated with the pyramidal morphology of the flakes were observed in our junction. In our initial submitted manuscript we considered this as an established fact because of the work of ref [36], where the authors indeed observed AB oscillations in samples with the same pyramidal morphology as our Bi_2Te_3 . The extensive experimental work we have done to answer the many questions of ref#2 has therefore led to various add-ons to the main text and to the supplementary material. We totally agree and understand the comments of referee # 5 but at this point, we would prefer to leave the discussion about the scattering mechanism as is and not to make major modifications to the text. In the end it strengthen the paper demonstrating the existence of at least one type of scattering mechanism, which can lead to "0" and " π " path. As rightly pointed out by referee #5 we have rephrased the final sentence of the discussion of the scattering mechanism in our devices as follow:

"Finally we would like to point out that magneto transport measurements (Supplementary Information, section IV) have clearly shown that scattering mechanisms, connected to the morphology of the Bi_2Te_3 flakes, take place in the junction channel, which is instrumental to get π trajectories. However the interpretation of our measurements does not change if other scattering mechanisms are also involved".

Referee #5

In the literature, there are some confusing statements about the symmetry of the induced order parameter in TI surfaces states; e.g., in ref. [1], it is stated that it "resembles a spinless $p_x + ip_y$ superconductor, but does not break time reversal symmetry." Other papers describe a helical p-wave (or mixed) order parameter symmetry. Here, the authors suggest a standard chiral p-wave order parameter but assert that the experimental signatures (temperature dependence of I_c , Fraunhofer pattern, etc.) are identical to s-wave superconductivity. Can the authors comment on this? Does the proposed mechanism for creating π channels (or domains) also work for other proposed order parameter symmetries, e.g., helical p-wave superconductivity?

The referee is again right: there is confusion in literature. In the original paper by Fu and Kane [ref 1] the order parameter induced in the TI surface states acquires a chiral $px + ipy$ form because of the very peculiar operator base the authors use, where the electron and holes operators acquire a phase factor. The chiral $px+ipy$ does not break time reversal symmetry in this particular operator base. In the same base Potter and Fu [Phys. Rev. B **88**, 121109(R) (2013), ref 23 of the revised manuscript] have calculated the magnetic pattern and have shown an almost ideal Fraunhofer-like pattern except for the nodes that are lifted by a very small amount $\Delta / \Phi_0 \approx 10$ nA in case of Al electrodes. The lifting is too tiny to be used as smoking gun for addressing Majoranas in the junction, and more importantly such a lifting of the nodes can be equally well emulated by structural non-homogeneity within the barrier (see textbook Barone Paterno for example). This is the reason why it is widely assumed that the magnetic pattern obtained with an induced OP in the TI surface states is Fraunhofer-like, analogously to an s-wave OP. The same applies to the $I_c(T)$.

Now it is worth pointing out that even though the operator base, introduced by Fu and Kane, is extremely elegant and gives a “simple” chiral p-wave (which does not break time reversal symmetry because of the phase factor of the quasiparticle operators) it is not intuitive and convenient to use. It gives an Andreev level picture, to describe the Josephson effect, very different from what we are used to deal with.

The derivation of the induced order parameter in the TI surface states by using a conventional base, where electron and holes are represented by canonical operators, has been derived by G. Tkachov et al. [PRB **88**, 075401 (2013), ref 14 of the revised manuscript]. The authors have derived an order parameter that they indicate as $p+s$. However the form of this OP is rather complex and consists of three orbital terms: $px+ipy$ (with a spin up-up configuration), $px-ipy$ (with a spin down-down configuration), s (with a singlet spin configuration). Since all the three components have different spin states they can be summed up only by using the Pauli matrices (note that because of the two chiral terms, time reversal symmetry is preserved as in Fu and Kane). In a conventional electron operator base it is not fully correct to represent the order parameter induced in the topological surface states as a single chiral term, because it is a mixture of $s+p$ (summed through Pauli Matrices).

From an experimental point of view, we believe it is advisable to consider G. Tkachov et al. formulation, where chiral components of the

p+s OP are the conventional chiral $px+ipy$ and $px-ipy$ (whose combination preserve time reversal symmetry) and we know how to represent them.

In our paper we have considered only the chiral part of the OP because it is the only component, which can be affected by a change of interface transparency. The s-wave component remains unaltered.

We have to acknowledge that we have not been clear enough on these crucial points. We have therefore made several changes throughout the main text to clarify statements that could bring to confusion and we have clarified our assumptions. All the made changes are highlighted in red.

Finally an helical $px\pm py$ is also compatible with a magnetic pattern with a minimum at zero magnetic field even though it does not come directly, as a possible symmetry, from the pioneering work of Fu and Kane as well as from the subsequent reformulation of G. Tkachov et al.. To answer to the referee comment we have added the following sentence:

“Clearly the dip at $B=0$ in the magnetic pattern is also compatible with a $d_{x^2-y^2}$ wave order parameter⁴¹. We can rule out this possibility since the $d_{x^2-y^2}$ OP does not change symmetry depending on the interface transparency³⁷ and therefore cannot account for the dramatic modifications of the magnetic field response of the junctions upon thermal cycling. An helical $px+py$ OP can instead be compatible with our entire experimental scenario. However this OP symmetry does not come directly from the theoretical works^{1,14,41} so the possibility to induce an helical $px+py$ OP needs further theoretical assessment.”

We now answer to the first question of referee #5.

Referee #5

It is widely believed that processes such as Ar ion milling create additional defects and thus increase (electronic) doping of the affected area. Also, the presence of a metallic film will likely cause band bending in the material underneath the electrodes. Therefore, I would expect some bulk doping in the S' region of the device. Does it make sense to assume that the induced order parameter has p-wave symmetry? A scenario with perhaps mixed effective order parameter (s- and p-wave) seems more likely to me. Does the proposed mechanism of creating an effective py order parameter after thermal

cycling work when the bulk conducts?

As we have discussed earlier the symmetry of the induced OP in the surface states is a mixed p+s, and we have clarified this issue throughout the main text. It is also true that there will be induced s-wave superconductivity in the bulk. However as already answered to referee #2 this circumstance will neither affect the formation of an effective p_y component at the interface (due to the reduced transparency after cycling) nor the interpretation of our measurements. Below we copy and paste the answer to this issue already given to referee #2.

“We do not exclude a contribution of the bulk to the transport properties of our junctions; however we cannot see how an eventual Josephson transport through the bulk with an s-wave induced order parameter, rather than a chiral p-wave, can change the interpretation of our measurement. The system with a Josephson current through the bulk can be considered as a parallel of a p+s OP and an s-wave OP at the first cool down and (knowing that only the p-wave component is affected by the low transparency of the interface) an effective p_y -wave and s-wave at the second cool down. Since (p+s)wave/TI/ (p+s)-wave junctions in magnetic field behave as s-wave junctions, at the first cool down, we observe a conventional Fraunhofer pattern.

Upon compressive strain (second cool down) we would have instead a p_y /TI/ p_y junction, with scattering events in the TI channel, in parallel with the s-wave channel from the bulk.

Cooper pair transport through the bulk would just act as the “0” trajectories in the p_y /TI/ p_y junctions effectively lifting the value of the Josephson current at zero external magnetic field, however leaving unaltered the dip structure at zero field of the magnetic pattern. We are not aware of any physical effects which could make the Josephson bulk transport either emulating a magnetic pattern, with a dip at zero externally applied field, or destroying such a feature”.

To clarify these points, we have added the following sentence to the revised main text:

“To conclude it is worth discussing if the interpretation we have provided of our experiment would change if we would also consider a parallel Josephson contribution due to the transport through the bulk with an s-wave OP. Cooper pair transport through the bulk would just act as “0” trajectories effectively lifting the value of the Josephson current at zero external magnetic field, however leaving unaltered the

dip structure at zero field of the magnetic pattern."

REVIEWERS' COMMENTS:

Reviewer #5 (Remarks to the Author):

In my opinion, the authors have significantly improved the manuscript. The authors' comments were helpful in explaining unclear points, especially, those regarding the order parameter on the topological surface. I believe, modeling of a more realistic critical current distribution has strengthened the argument for the existence of "0" and " π " channels in the weak link. Regardless, the physical mechanism of creating such " π " channels is – and remains – speculative as the transmission characteristics of the weak link as such are unknown after deformation.

I think the experimental work deserves publication in a journal with wide readership. It is likely to inspire theoretical efforts to improve our understanding of scattering mechanisms in materials with topological band structure.